# Online Learning of Quantum States with Logarithmic Loss via VB-FTRL

**Wei-Fu Tseng**                                                            WEIFU.TSENG03@GMAIL.COM
*Department of Mathematics, National Taiwan University*

**Kai-Chun Chen**                                                          CASPER901208@GMAIL.COM
*Department of Electrical Engineering, National Taiwan University*
*Graduate Institute of Communication Engineering, National Taiwan University*

**Zi-Hong Xiao**                                                                  ERIC@XIAO.TW
*School of Medicine, National Taiwan University*

**Yen-Huan Li**                                                    YENHUAN.LI@CSIE.NTU.EDU.TW
*Department of Computer Science and Information Engineering, National Taiwan University*
*Department of Mathematics, National Taiwan University*
*Center for Quantum Science and Engineering, National Taiwan University*

**Editors:** Gautam Kamath and Po-Ling Loh

## Abstract

Online learning of quantum states with the logarithmic loss (LL-OLQS) is a quantum generalization of online portfolio selection (OPS), a classic open problem in online learning for over three decades. This problem also emerges in designing stochastic optimization algorithms for maximum-likelihood quantum state tomography. Recently, Jézéquel et al. (2022, arXiv:2209.13932) proposed the VB-FTRL algorithm, the first regret-optimal algorithm for OPS with moderate computational complexity. In this paper, we generalize VB-FTRL for LL-OLQS. Let $d$ denote the dimension and $T$ the number of rounds. The generalized algorithm achieves a regret rate of $O(d^2 \log(d + T))$ for LL-OLQS. Each iteration of the algorithm consists of solving a semidefinite program that can be implemented in polynomial time by, for example, cutting-plane methods. For comparison, the best-known regret rate for LL-OLQS is currently $O(d^2 \log T)$, achieved by an exponential weight method. However, no explicit implementation is available for the exponential weight method for LL-OLQS. To facilitate the generalization, we introduce the notion of VB-convexity. VB-convexity is a sufficient condition for the volumetric barrier associated with any function to be convex and is of independent interest.

**Keywords:** Online learning of quantum states, VB-FTRL, logarithmic regret, volumetric barrier.

## 1. Introduction

A quantum state is characterized by a *density matrix*, which is a Hermitian positive semi-definite matrix with unit trace. Denote by $\mathcal{D}_d$ the set of density matrices in $\mathbb{C}^{d \times d}$. Consider the following sequential game between two strategic players, say Physicist and Reality:

- There are in total $T$ rounds.

- In the $t$-th round,

    1. First, Physicist announces a density matrix $\rho_t \in \mathcal{D}_d$;
    2. then, Reality announces a Hermitian positive semi-definite matrix $A_t \in \mathbb{C}^{d \times d}$;

3. finally, Physicist suffers a loss of value $f_t(\rho_t)$, where the loss function $f_t$ is given by $f_t(\rho) := -\log \operatorname{tr}(A_t\rho)$.

- The goal of Physicist is to attain a regret that is as small as possible against all possible strategies of Reality. The regret is given by

$$\operatorname{Regret}_T := \sum_{t=1}^{T} f_t(\rho_t) - \min_{\rho \in \mathcal{D}_d} \sum_{t=1}^{T} f_t(\rho).$$

The game arises in the design of efficient stochastic optimization algorithms for maximum-likelihood quantum state tomography and approximating the permanents of positive semi-definite matrices (Lin et al., 2021; Tsai et al., 2024). It is also an instance of the problem of online learning of quantum states proposed by Aaronson et al. (2018, 2019). For convenience, we will refer to the game as "online learning of quantum states with the logarithmic loss," abbreviated as LL-OLQS.

In their seminal work, Aaronson et al. (2018, 2019) considered the absolute loss and suggested studying online learning of quantum states with other loss functions as a direction for future research. Assuming that the loss functions are convex and Lipschitz continuous, existing results in the literature have demonstrated that standard online convex optimization algorithms and their regret guarantees readily apply to the problem of online learning of quantum states (Aaronson et al., 2018, 2019; Bansal et al., 2024; Chen et al., 2022; Yang et al., 2020)[1]. Unfortunately, it is easily verified that the logarithmic losses in LL-OLQS violate the Lipschitz continuity assumption, so standard results in online convex optimization (Hazan, 2023; Orabona, 2023) do not directly apply.

To illustrate the challenges of addressing LL-OLQS, consider its *classical* counterpart, where all matrices involved share a common eigenbasis. In this case, rather than working directly with the matrices, it suffices to consider the vectors of their eigenvalues. The set $\mathcal{D}_d$ can be replaced with the probability simplex $\Delta_d$ in $\mathbb{R}^d$. The outputs of Physicist and Reality can then be replaced with $x_t \in \Delta_d$ and $a_t \in [0, +\infty)^d$, respectively. The loss functions become $f_t(x) := -\log \langle a_t, x_t \rangle$. As noted by Lin et al. (2021) and Zimmert et al. (2022), this classical formulation corresponds to the problem of online portfolio selection.

Designing an online portfolio selection algorithm—optimal in both regret and computational complexity—has remained a classic open problem for over 30 years. The interested reader is referred to, for example, the discussions by van Erven et al. (2020) and Jézéquel et al. (2022), along with the references therein. While the optimal regret for online portfolio selection is known to be $O(d \log T)$, the sole algorithm known to achieve this optimal regret, Universal Portfolio (Cover, 1991; Cover and Ordentlich, 1996), suffers from a high per-round computational complexity of $O(d^4 T^{13})$ (Kalai and Vempala, 2002). On the other hand, the lowest per-round computational complexity achievable, which is $O(d)$, is met by several existing algorithms (Helmbold et al., 1998; Nesterov, 2011; Orseau et al., 2017; Tsai et al., 2023a,b); however, these algorithms do not achieve logarithmic (in $T$) regret rates.

Due to the non-commutativity nature of the quantum setup, LL-OLQS is even more challenging than online portfolio selection. For LL-OLQS, the optimal regret has remained unclear. The best-known regret rate for LL-OLQS is $O(d^2 \log T)$, achieved using a direct extension of Universal

---

1. Notice that quantum density matrices are complex; this introduces subtle issues in analyzing online convex optimization algorithms. Most of these works base their novelty primarily on handling complex variables. We show in Appendix A that such additional efforts are not necessary.

Portfolio (Zimmert et al., 2022). However, this algorithm requires evaluating a high-dimensional integral in each round, and no explicit implementation is currently available. The Schrodinger's BISONS algorithm achieves a regret rate of $O(d^3 \log^2(T))$ with a per-round computational complexity of $\tilde{O}(\text{poly}(d))$ (Zimmert et al., 2022). The Q-Soft-Bayes and Q-LB-OMD algorithms achieve a regret rate of $\tilde{O}(\sqrt{dT})$ with a significantly lower per-round computational complexity of $\tilde{O}(d^3)$ (Lin et al., 2021; Tsai et al., 2023a). These three trade-offs between regret and computational efficiency represent the current "regret-efficiency frontier" of existing algorithms. In other words, achieving a lower regret rate necessitates a higher computational complexity, and vice versa.

Recently, Jézéquel et al. (2022) proposed an efficient and regret-optimal algorithm for OPS, named VB-FTRL. This algorithm achieves a regret rate of $O(d \log(d + T))$ with a per-round computational complexity of $\tilde{O}(d^2(T + d))$. Compared to the regret-optimal Universal portfolio algorithm, VB-FTRL achieves the same regret rate when $T > d$ but with significantly lower per-round computational complexity. This prompts the question: Can VB-FTRL be generalized for LL-OLQS?

In this paper, we prove the following theorem.

**Theorem 1** *There is an algorithm that achieves a regret rate of $O(d^2 \log(d + T))$ for LL-OLQS. Each iteration of the algorithm consists of solving a semidefinite program.*

Notice that semidefinite programs can be solved in polynomial time by, for example, cutting-plane methods (Lee and Vempala, 2025, Chapter 3). We provide an implementation of the algorithm in Appendix F, which uses a cutting-plane method and has a per-round computational complexity of $O(Td^8 + d^{10})$. Jézéquel et al. (2022) designed a quasi-Newton method to implement VB-FTRL with a per-round computational complexity of $\tilde{O}(d^2(T + d))$, significantly faster than cutting-plane methods. We leave generalization of the quasi-Newton method for the quantum setup as a future research direction.

Two challenges emerge in proving Theorem 1. The first challenge is addressing the affine reparameterization step in the regret analyses for both LL-OLQS (Jézéquel et al., 2022) and its classical counterpart, online portfolio selection (van Erven et al., 2020). The affine reparameterization step enables one to directly apply existing results for self-concordant functions (Nesterov, 2018). Unfortunately, it does not extend to the quantum setup. We adapt the proof strategy of Tsai et al. (2023b) to our analysis to bypass the affine reparameterization step.

The second challenge lies in verifying convexity of the volumetric barrier (VB) associated with the regularized cumulative loss in our algorithm. Jézéquel et al. (2022) accomplished a similar verification for the classical case. Their explicit calculations, nevertheless, appear to be challenging in the quantum case. Generalizations of the VB for semidefinite programming, or the quantum setup from our perspective, have been studied by Nesterov and Nemirovskii (1994) and Anstreicher (2000), but their assumptions do not hold in our case. To this end, we introduce the notion of *VB-convexity* and prove that the VB associated with any VB-convex function is necessarily convex. Then, we verify that the regularized cumulative loss in our algorithm is actually VB-convex. The notion of VB-convexity is of independent interest.

We conclude the introduction with a discussion on the optimality of Theorem 1. Unlike in online portfolio selection, the minimax optimal regret rate for LL-OLQS remains unclear. While the Universal Portfolio algorithm is known to be regret-optimal for online portfolio selection, to the best of our knowledge, no optimality guarantee has been proven for the $O(d^2 \log T)$ regret rate achieved by its quantum generalization. Note that there is a gap of $d$ between the regret rate of

the Universal Portfolio, which is $O(d \log T)$, and that of its quantum generalization. Therefore, the optimality of the former does not imply the optimality of the latter. We suspect that a regret rate of $O(d^2 \log T)$ is minimax optimal for LL-OLQS. If so, the regret rate of the proposed algorithm, which will be detailed in Section F, would also be minimax optimal when $T > d$. Characterizing the minimax regret rate would be an essential next step following this work.

## 1.1. Notation

As density matrices are complex, defining notions like $\nabla f, \nabla^2 f$ can be tricky. The reader is referred to Section A for formal definitions and discussions.

For any complex matrix $M$, we denote its transpose by $M^\mathsf{T}$ and its conjugate transpose by $M^*$. We will use $I$ to represent the identity matrix. Let $\mathcal{H}^d$ denote the set of Hermitian matrices in $\mathbb{C}^{d \times d}$. For any $A, B \in \mathcal{H}^d$, we write $A \geq B$ if the matrix $A - B$ is positive semi-definite. We use $A \otimes B$ to denote the Kronecker product of $A$ and $B$.

For any matrix $M \in \mathcal{H}^d$, we write $\mathrm{vec}(M)$ for the vectorization of $M$, i.e., the vector formed by stacking the columns of the matrix $M$ on top of one another. For any function $\varphi : \mathcal{H}^d \subseteq \mathbb{C}^{d \times d} \to \mathbb{R}$, we write $\overline{\varphi} : \mathbb{C}^{d^2} \to \mathbb{R}$ for the function $\overline{\varphi}$ such that $\varphi(M) = \overline{\varphi}(\mathrm{vec}(M))$ for any $M \in \mathcal{H}^d$. Therefore, for example, $\nabla \overline{\varphi}(\mathrm{vec}(M))$ is a $d^2$-dimensional vector and $\nabla^2 \overline{\varphi}(\mathrm{vec}(M))$ is a $d^2 \times d^2$ matrix. For convenience, we write $\nabla^{-2} f$ for the inverse of the Hessian of the function $f$.

For any vector $v \in \mathbb{C}^d$ and Hermitian positive definite matrix $A \in \mathbb{C}^{d \times d}$, we write $\|v\|_A$ for the norm defined by $A$, i.e., $\|v\|_A := \sqrt{\langle v, Av \rangle}$, where $\langle v, Av \rangle = v^* A v$ and $v^*$ denotes the conjugate transpose of $v$.

## 2. VB-Convex Functions

The notion of a VB was originally defined with respect to a polytope (Vaidya, 1996). For any polytope, let $\mathrm{LB}(x)$ denote the associated logarithmic barrier; the VB is given by the function $(1/2) \log \det \nabla^2 \mathrm{LB}(x)$. For convenience of presentation, we will refer to "the VB associated with a function $f(x)$" as the function $(1/2) \log \det \nabla^2 f(x)$ and denote it by $\mathrm{VB}_f(x)$ throughout this paper.

To motivate the notion of VB-convexity, let us start by reviewing the relevant part in the analysis of VB-FTRL by Jézéquel et al. (2022) for online portfolio selection. Online portfolio selection is an online convex optimization problem in which the constraint set is the probability simplex in $\mathbb{R}^d$, and the loss function for the $t$-th round is given by $-\log \langle a_t, x \rangle$, where $a_t$ are adversarially chosen entry-wise non-negative vectors. Consider the function

$$\ell_t(x) := -\sum_{\tau=1}^{t} \log \langle a_\tau, x \rangle - \sum_{i=1}^{d} \log x[i],$$

where $x[i]$ denotes the $i$-th entry of the vector $x$; the function is the cumulative loss regularized by the Burg entropy. As the probability simplex is a $(d-1)$-dimensional object, Jézéquel et al. (2022) introduced an affine transformation $A : \mathbb{R}^{d-1} \to \mathbb{R}^d$ and considered the function $\tilde{\ell}_t(u) := \ell_t(Au)$. The regret analysis of VB-FTRL requires the function $\mathrm{VB}_{\tilde{\ell}_t}(u)$ to be convex. Notice that the function $\tilde{\ell}_t$ is indeed the logarithmic barrier of a $(d-1)$-dimensional polytope. The convexity

of $\mathrm{VB}_{\ell_t}(x)$ immediately follows from the following lemma (Vaidya, 1996) (see also the discussion by Jézéquel et al. (2022, Lemma C.4) [2].

**Lemma 2 (Vaidya (1996, Lemma 3))** *Let $\mathrm{LB}(x)$ be the logarithmic barrier associated with a full-dimensional polytope. Then, it holds that $Q(x) \leq \nabla^2 \mathrm{VB}_{\mathrm{LB}}(x) \leq 5Q(x)$ for some positive definite matrix-valued function $Q(x)$.*

A natural generalization of the Burg entropy for density matrices is the negative log-determinant function

$$R(\rho) := -\log \det \rho, \quad \forall \rho \in \mathcal{D}_d. \tag{1}$$

Define

$$L_t(\rho) := \sum_{\tau=1}^{t} f_\tau(\rho) + \lambda R(\rho), \quad \forall t \in \mathbb{N},$$

where $f_\tau(\rho) = -\log \mathrm{tr}(A_\tau \rho)$ is the loss function in LL-OLQS. Our analysis will require the convexity of the VB associated with the function $\overline{L}_t$; note that we have introduced vectorization here.

Whereas there are generalizations of VB for semi-definite programming, existing results do not directly apply. Nesterov and Nemirovskii (1994) proved that the VB associated with the function $-\log \det(\mathcal{A}\cdot)$ is self-concordant for any linear mapping $\mathcal{A} : \mathbb{R}^n \to \mathbb{R}^{d \times d}$ under the assumption that $d > n$; Anstreicher (2000) proved that the VB associated with the function $-\log \det S(\cdot)$ is self-concordant for any affine mapping $S(x) := \sum_{i=1}^{d^2} x(i) A_i - C : \mathbb{R}^{d^2} \to \mathbb{R}^{d \times d}$ under the assumptions that $A_i$ and $C$ are symmetric matrices. It is easily verified that both assumptions fail for $\mathrm{VB}_{\overline{R}}$.

To this end, we introduce the notion of *VB-convexity*. We prove that $\mathrm{VB}_\varphi$ is necessarily convex for any VB-convex function $\varphi$; moreover, VB-convexity is closed under addition. Then, we verify that the negative log-determinant function (1) and the loss functions in LL-OLQS (Section 1) are indeed VB-convex. These results imply the VB-convexity of the function $\overline{L}_t$ (recall that $L_t(\cdot) = \overline{L}_t(\mathrm{vec}(\cdot))$), which has two implications. First, this implies that the algorithm in Theorem 1 only needs to solve a convex optimization problem in each iteration, which can be implemented by cutting-plane methods as described in Appendix F. Second, the convexity of the volumetric barrier is utilized in Section 3.2.3 to analyze the regret, similarly to what is done in the proof of Jézéquel et al. (2022) .

### 2.1. General Theory

Below we present the definition of VB-convexity. It is easily to check that the negative logarithmic function $-\log x$ satisfies the following definition with equality.

**Definition 3 (VB-Convexity)** *Let $\mathbb{H}$ be a real Hilbert space. Consider a function $\varphi \colon \mathbb{H} \to \mathbb{R}$ such that $\varphi \in C^4(\mathrm{dom}\,\varphi)$. We say that $\varphi$ is VB-convex if it is strictly convex and*

$$D^4\varphi(x)[u,u,v,v]D^2\varphi(x)[v,v] \geq \frac{3}{2}\left(D^3\varphi(x)[u,v,v]\right)^2, \quad \forall x \in \mathrm{dom}\,\varphi, u, v \in \mathbb{H}.$$

---

2. To be precise, Jézéquel et al. (2022) considered a function that slightly differs from the $\ell_t(x)$ defined here. They considered a version where weights are applied to $\sum_{i=1}^{d} \log x[i]$. As a result, additional work is required to adjust the proof of Vaidya's lemma. However, the modification is relatively straightforward and it is unnecessary to develop a new theory such as our VB-convexity.

Below are two useful properties of VB-convex functions. The first is immediate from the definition. The proof for the second can be found in Section D.

**Lemma 4**  *VB-convexity is affine invariant. That is, if a function $\varphi(x)$ is VB-convex, then the function $\varphi(Ax)$ is also VB-convex for any affine operator $A$.*

**Lemma 5**  *Let $\varphi_1$ and $\varphi_2$ be VB-convex functions and $\alpha, \beta > 0$. Then, the function $\varphi := \alpha\varphi_1 + \beta\varphi_2$ is also a VB-convex function.*

These two properties together with Definition 3 are reminiscent of self-concordant functions (see Definition 18 and Lemma 20 ). Definition 3 is indeed inspired by the definition of self-concordant functions. However, we were unable to identify any clear relation between self-concordance and VB-convexity.

We now proceed to prove the convexity of the VB associated with any VB-convex function (Corollary 8). The following technical lemma is necessary for its proof, whose proof can be found in Appendix D.

**Lemma 6**  *Let $A$, $B$, and $C$ be $d \times d$ Hermitian matrices. Suppose that $A$ is positive semi-definite, $B$ is positive definite, and*

$$\langle v, Av \rangle \langle v, Bv \rangle \geq \langle v, Cv \rangle^2, \quad \forall v \in \mathbb{C}^d.$$

*Then, it holds that*

$$\operatorname{tr}\bigl(AB^{-1}\bigr) \geq \operatorname{tr}\bigl(B^{-1}CB^{-1}C\bigr).$$

**Theorem 7**  *Let $\varphi$ be an VB-convex function. Define the associated volumetric barrier as*

$$\operatorname{VB}_\varphi(x) := \frac{1}{2} \log \det \nabla^2 \varphi(x).$$

*Let $Q(x)$ be a Hermitian positive semi-definite matrix defined via the equality*

$$\langle u, Q(x)u \rangle = \frac{1}{2} \operatorname{tr}\left( \nabla^{-2}\varphi(x) D^4\varphi(x)[u, u] \right), \quad \forall u \in \mathbb{C}^d.$$

*Then, it holds that*

$$\frac{1}{3}Q(x) \leq \nabla^2 \operatorname{VB}_\varphi(x) \leq Q(x), \quad \forall x \in \operatorname{dom} \varphi. \tag{2}$$

**Proof**  Let $S(x)$ be a Hermitian matrix defined via the equality

$$\langle u, S(x)u \rangle = \frac{1}{2} \operatorname{tr}\left( \nabla^{-2}\varphi(x) D^3\varphi(x)[u] \nabla^{-2}\varphi(x) D^3\varphi(x)[u] \right), \quad \forall u \in \mathcal{H}^d.$$

Then, a direct calculation using the chain rule (Lemma 17) gives

$$D \operatorname{VB}_\varphi(x)[u] = \frac{1}{2} \operatorname{tr}\bigl(\nabla^{-2}\varphi(x) D^3\varphi(x)[u]\bigr),$$
$$D^2 \operatorname{VB}_\varphi(x)[u, u] = \langle u, Q(x)u \rangle - \langle u, S(x)u \rangle.$$

We now check the positive definiteness of the matrices $Q(x)$ and $S(x)$ by using the fact that $\mathrm{tr}(AB) \geq 0$ for Hermitian positive semi-definite matrices $A$ and $B$. By Definition 3, the matrix $\nabla^{-2}\varphi(x)$ is positive definite and $D^4\varphi(x)[u, u]$ is positive semi-definite. This implies that the matrix $Q$ is positive semi-definite. Given the positive definiteness of $\nabla^{-2}\varphi(x)$, the matrix $D^3\varphi(x)[u]\nabla^{-2}\varphi(x)D^3\varphi(x)[u]$ is also positive definite, ensuring that the matrix $S(x)$ is positive definite.

The right-hand side of the inequality (2) follows from $S(x)$ is positive definite. For the left-hand side, it suffices to prove that $Q(x) \geq (3/2)S(x)$, or equivalently,

$$\mathrm{tr}\left(\nabla^{-2}\varphi(x)D^4\varphi(x)[u, u]\right) \geq \frac{3}{2}\mathrm{tr}\left(\nabla^{-2}\varphi(x)D^3\varphi(x)[u]\nabla^{-2}\varphi(x)D^3\varphi(x)[u]\right).$$

Let $A = (2/3)D^4\varphi(x)[u, u]$, $B = \nabla^2\varphi(x)$, and $C = D^3\varphi(x)[u]$. Then, by the definition of VB-convexity (Definition 3), we have

$$\langle v, Av \rangle \langle v, Bv \rangle \geq \langle v, Cv \rangle^2, \quad \forall v \in \mathbb{C}^d.$$

The left-hand side of the inequality (2) follows from Lemma 6. ∎

The following corollary immediately follows since the matrix $Q(x)$ in Theorem 1 is positive semi-definite.

**Corollary 8** *The volumetric barrier associated with any VB-convex function is convex.*

## 2.2. VB-Convexity of $\overline{L}_t$

We show that the loss functions in LL-OLQS and the negative log-determinant function are both VB-convex.

**Lemma 9** *Let $f_t$ be the loss functions in the LL-OLQS game (Section 1). Then, the functions $\overline{f}_t$ are VB-convex.*

**Proof** It is easily verified that $\varphi(x) := -\log x$ is VB-convex. Then, the lemma follows from the affine invariance of VB-convexity (Lemma 4). ∎

We will use the following lemma to establish the VB-convexity of the log-determinant function in Lemma 11.

**Lemma 10** *For any $n \times n$ Hermitian matrices $A$ and $B$, it holds that*

$$\mathrm{tr}\left(A^2B^2\right) \geq \mathrm{tr}(ABAB).$$

**Proof** Since $A$ and $B$ are Hermitian, the matrices $ABA$, $B$, $A^2$, and $B^2$ are Hermitian. Therefore, both $\mathrm{tr}(ABAB)$ and $\mathrm{tr}\left(A^2B^2\right)$ are real numbers. Notice that for any skew-Hermitian matrix $M$, we have $\mathrm{tr}\left(M^2\right) = -\langle M, M \rangle_{\mathrm{HS}} \leq 0$. Taking $M = AB - BA$, which is obviously skew-Hermitian, we write

$$0 \geq \frac{1}{2}\mathrm{tr}\left((AB - BA)^2\right) = \mathrm{tr}(ABAB) - \mathrm{tr}\left(A^2B^2\right).$$

∎

**Lemma 11** *Denote by $R$ the negative log-determinant function, i.e., $R(\rho) := -\log \det \rho$. Then, the function $\overline{R}$ is VB-convex.*

**Proof** We aim to verify the inequality

$$D^4\overline{R}(\mathrm{vec}(\rho))[u, u, v, v] D^2\overline{R}(\mathrm{vec}(\rho))[v, v] \geq \frac{3}{2} \left(D^3\overline{R}(\mathrm{vec}(\rho))[u, v, v]\right)^2.$$

Define $U := \mathrm{vec}^{-1}(u)$ and $V := \mathrm{vec}^{-1}(v)$. By Lemma 24, we have

$$D^2\overline{R}(\mathrm{vec}(\rho))[v, v] = \mathrm{tr}\left(\rho^{-1}V\rho^{-1}V\right),$$
$$D^3\overline{R}(\mathrm{vec}(\rho))[u, v, v] = -2\,\mathrm{tr}\left(\rho^{-1}V\rho^{-1}V\rho^{-1}U\right),$$
$$D^4\overline{R}(\mathrm{vec}(\rho))[u, u, v, v] = 4\,\mathrm{tr}\left(\rho^{-1}U\rho^{-1}U\rho^{-1}V\rho^{-1}V\right) + 2\,\mathrm{tr}\left(\rho^{-1}U\rho^{-1}V\rho^{-1}U\rho^{-1}V\right).$$

Define $N := \rho^{-1}U\rho^{-1}V\rho^{-1} + \rho^{-1}V\rho^{-1}U\rho^{-1}$. Then, we have

$$\mathrm{tr}(NV) = 2\,\mathrm{tr}\left(\rho^{-1}V\rho^{-1}V\rho^{-1}U\right) = -D^3\overline{R}(\mathrm{vec}(\rho))[u, v, v].$$

Applying Lemma 10 with $A = \rho^{-1/2}U\rho^{-1/2}$ and $B = \rho^{-1/2}V\rho^{-1/2}$, we write

$$D^4\overline{R}(\mathrm{vec}(\rho))[u, u, v, v]$$
$$= \left[4\,\mathrm{tr}\left(\rho^{-1}U\rho^{-1}U\rho^{-1}V\rho^{-1}V\right) + 2\,\mathrm{tr}\left(\rho^{-1}U\rho^{-1}V\rho^{-1}U\rho^{-1}V\right)\right]$$
$$\geq \left[3\,\mathrm{tr}\left(\rho^{-1}U\rho^{-1}U\rho^{-1}V\rho^{-1}V\right) + 3\,\mathrm{tr}\left(\rho^{-1}U\rho^{-1}V\rho^{-1}U\rho^{-1}V\right)\right]$$
$$= \frac{3}{2}\,\mathrm{tr}\left(N\rho N\rho\right).$$

By the Cauchy-Schwarz inequality, we write

$$D^4\overline{R}(\mathrm{vec}(\rho))[u, u, v, v] D^2\overline{R}(\mathrm{vec}(\rho))[v, v]$$
$$= \frac{3}{2}\,\mathrm{tr}\left(N\rho N\rho\right)\mathrm{tr}\left(\rho^{-1}V\rho^{-1}V\right)$$
$$= \frac{3}{2}\,\langle\rho^{1/2}N\rho^{1/2}, \rho^{1/2}N\rho^{1/2}\rangle_{\mathrm{HS}}\,\langle\rho^{-1/2}V\rho^{-1/2}, \rho^{-1/2}V\rho^{-1/2}\rangle_{\mathrm{HS}}$$
$$\geq \frac{3}{2}\left(\langle\rho^{1/2}N\rho^{1/2}, \rho^{-1/2}V\rho^{-1/2}\rangle_{\mathrm{HS}}\right)^2$$
$$= \frac{3}{2}\,\mathrm{tr}(NV)^2$$
$$= \frac{3}{2}\left(D^3\overline{R}(\mathrm{vec}(\rho))[u, v, v]\right)^2.$$

∎

We conclude this section with the following corollary, which states that the function $\overline{L}_t$ is VB-convex, and hence the function $-\log\det\nabla^2\overline{L}_t(\cdot)$ is convex.

**Corollary 12** *The function $\overline{L}_t(\rho)$ is VB-convex.*

**Proof** Lemma 9 and Lemma 11 establish the VB-convexity of the functions $\overline{f}_t$ and $\overline{R}$, respectively. Since VB-convexity is closed under addition (Lemma 5), the function $\overline{L}_t = \sum_{\tau=1}^t \overline{f}_\tau + \overline{R}$ is also VB-convex. ∎

## 3. Proof of Theorem 1

### 3.1. Algorithm

The algorithm we propose is a direct generalization of VB-FTRL (Jézéquel et al., 2022) for LL-OLQS, except that it gets rid of the affine reparameterization step in VB-FTRL (see Section 3.2.3 for details). Notice that when all matrices involved share a common eigenbasis, our algorithm specializes to an OPS algorithm. Define $R(\rho) := -\log \det \rho$. The algorithm proceeds as follows:

- Let $P_0(\rho) := \lambda R(\rho)$ for some $\lambda > 0$.

- At the $t$-th round, the algorithm outputs

$$\rho_t \in \operatorname*{argmin}_{\rho \in \mathcal{D}_d} P_{t-1}(\rho),$$

  where

$$P_t(\rho) := L_t(\rho) + \mu V_t(\rho),$$

$$L_t(\rho) := \sum_{\tau=1}^{t} f_\tau(\rho) + \lambda R(\rho),$$

$$V_t(\rho) := \frac{1}{2} \log \det \nabla^2 \overline{L}_t(\operatorname{vec}(\rho)).$$

### 3.2. Regret Analysis

Our goal in this sub-section is to prove the following theorem, which implies Theorem 1.

**Theorem 13** *The algorithm in the previous sub-section achieves a regret rate of $O(d^2 \log(T + d))$ with $\lambda = 300$ and $\mu = 10$.*

The following regret analysis essentially follows the strategy of Jézéquel et al. (2022). As discussed in Section 1, two major differences are:

1. We avoid the affine reparametrization step.

2. We establish the convexity of $V_t$, the VB associated with $L_t$, via the approach in Section 2.

Notice that $P_0(\rho) = -\log \det(\rho) \geq 0$ by definition. We decompose the regret as

$$
\begin{aligned}
\operatorname{Regret}_T &= \sum_{t=1}^{T} f_t(\rho_t) - \min_{\rho \in \mathcal{D}_d} \sum_{t=1}^{T} f_t(\rho) \\
&\leq \sum_{t=1}^{T} \left( f_t(\rho_t) + \min_{\rho \in \mathcal{D}_d} P_{t-1}(\rho) - \min_{\rho \in \mathcal{D}_d} P_t(\rho) \right) - \min_{\rho \in \mathcal{D}_d} \sum_{t=1}^{T} f_t(\rho) + \min_{\rho \in \mathcal{D}_d} P_T(\rho) \\
&= \sum_{t=1}^{T} \left( P_t(\rho_t) + \mu V_{t-1}(\rho_t) - \mu V_t(\rho_t) - P_t(\rho_{t+1}) \right) - \min_{\rho \in \mathcal{D}_d} \sum_{t=1}^{T} f_t(\rho) + \min_{\rho \in \mathcal{D}_d} P_T(\rho).
\end{aligned}
$$

Then, we have

$$\text{Regret}_T \leq \text{Bias}_T + \sum_{t=1}^{T} (\text{Gain}_t + \text{Miss}_t),$$

where

$$\text{Bias}_T := \min_{\rho \in \mathcal{D}_d} P_T(\rho) - \min_{\rho \in \mathcal{D}_d} \sum_{t=1}^{T} f_t(\rho),$$

$$\text{Gain}_t := \mu \left( V_{t-1}(\rho_t) - V_t(\rho_t) \right), \quad \text{and}$$

$$\text{Miss}_t := P_t(\rho_t) - P_t(\rho_{t+1}).$$

The term $\text{Bias}_T$ represents the bias of using the volumetric barrier and the negative log-determinant function as regularizers. The term $\text{Gain}_t$ measures the benefit of introducing the volumetric barrier $V_t$, which depends on the number of rounds $t$. The term $\text{Miss}_t$ arises because the algorithm does not have access to the loss function $f_t$ at the $t$-th round; that is, the algorithm can only follow the regularized leader rather than be the regularized leader. Define

$$H_t(\rho) := \nabla^2 \overline{L}_t(\text{vec}(\rho)), \quad \pi_t(\rho) := \left\| \nabla \overline{f}_t(\text{vec}(\rho)) \right\|_{[H_t(\rho)]^{-1}}^2,$$

$$H_t := H_t(\rho_t), \qquad\qquad \pi_t := \pi_t(\rho_t).$$

The rest of analysis, to be presented in the following three subsections, consists of three parts.

1. We prove that $\text{Bias}_T = O(d^2 \log(T + d))$.

2. We prove that $\text{Gain}_t \leq -(\mu/2)\pi_t$.

3. We prove that $\text{Miss}_t \leq (\mu/2)\pi_t$.

The three results together imply the regret bound in Theorem 13.

### 3.2.1. BOUNDING $\text{Bias}_T$

We start with the decomposition $\text{Bias}_T = \text{VolBias}_T + \text{LogBias}_T$, where

$$\text{VolBias}_T := \min_{\rho \in \mathcal{D}_d} P_t(\rho) - \min_{\rho \in \mathcal{D}_d} L_t(\rho), \quad \text{LogBias}_T := \min_{\rho \in \mathcal{D}_d} L_t(\rho) - \min_{\rho \in \mathcal{D}_d} \sum_{t=1}^{T} f_t(\rho).$$

Here, $\text{VolBias}_T$ and $\text{LogBias}_T$ are the biases due to the volumetric barrier and the negative log-determinant function, respectively.

We first bound $\text{VolBias}_T$. Let $\rho_T^\star \in \text{argmin}_{\rho \in \mathcal{D}_d} L_T(\rho)$. Then, we write

$$\text{VolBias}_T \leq P_t(\rho_T^\star) - L_t(\rho_T^\star)$$
$$= \mu V_T(\rho_T^\star)$$
$$= \frac{\mu}{2} \log \det \nabla^2 \overline{L}_t(\text{vec}(\rho_T^\star)).$$

The function $\overline{L}_t$ is strictly convex so $\lambda_{\max}(\nabla^2\overline{L}_t(\text{vec}(\rho_T^\star)))$, the largest eigenvalue of its Hessian, is strictly positive. We obtain

$$\text{VolBias}_T \leq \frac{\mu d^2}{2}\log\lambda_{\max}(\nabla^2\overline{L}_t(\text{vec}(\rho_T^\star))). \tag{3}$$

By Lemma 27, Lemma 29 and Lemma 30, we write

$$\begin{aligned}
\nabla^2\overline{L}_t(\text{vec}(\rho_T^\star)) &= \sum_{t=1}^{T}\nabla^2\overline{f}_t(\text{vec}(\rho_T^\star)) + \lambda\nabla^2\overline{R}(\text{vec}(\rho_T^\star)) \\
&\leq (T+\lambda)\,\nabla^2\overline{R}(\text{vec}(\rho_T^\star)) \\
&= (T+\lambda)\left(((\rho_T^\star)^{-1})^{\mathsf{T}}\otimes(\rho_T^\star)^{-1}\right) \\
&\leq \frac{(T+\lambda d)^3}{\lambda^2}(I\otimes I).
\end{aligned}$$

Therefore,

$$\text{VolBias}_T \leq \frac{\mu d^2}{2}\log\frac{(T+\lambda d)^3}{\lambda^2} = O(d^2\log(T+d)). \tag{4}$$

We then bound $\text{LogBias}_T$. Define

$$\rho_T^\circ := \operatorname*{argmin}_{\rho\in\mathcal{D}_d}\sum_{t=1}^{T}f_t(\rho), \quad [\rho_T^\circ]_\alpha := (1-\alpha)\rho^\circ + \frac{\alpha}{d}I.$$

By Lemma 31, we write

$$\begin{aligned}
\text{LogBias}_T &= \min_{\rho\in\mathcal{D}_d}L_t(\rho) - \min_{\rho\in\mathcal{D}_d}\sum_{t=1}^{T}f_t(\rho) \\
&\leq L_t([\rho_T^\circ]_\alpha) - \sum_{t=1}^{T}f_t(\rho_T^\circ) \\
&= L_t([\rho_T^\circ]_\alpha) - \sum_{t=1}^{T}f_t([\rho_T^\circ]_\alpha) + \sum_{t=1}^{T}f_t([\rho_T^\circ]_\alpha) - \sum_{t=1}^{T}f_t(\rho_T^\circ) \\
&\leq \lambda R([\rho_T^\circ]_\alpha) + \frac{\alpha}{1-\alpha}T \\
&\leq -\lambda d\log\left(\frac{\alpha}{d}\right) + \frac{\alpha}{1-\alpha}T.
\end{aligned}$$

Let $\alpha = \lambda d/(T+\lambda d)$. We obtain

$$\text{LogBias}_T \leq -\lambda d\log\left(\frac{\lambda}{T+\lambda d}\right) + \lambda d = O(d\log(T+d)). \tag{5}$$

Combining the two inequalities (4) and (5), we conclude that $\text{Bias}_T = O(d^2\log(T+d))$.

### 3.2.2. BOUNDING $\mathrm{Gain}_t$

By the fact that $\det(AB) = \det(A)\det(B)$, we write

$$
\begin{aligned}
\mathrm{Gain}_t &= \mu\left(V_{t-1}(\rho_t) - V_t(\rho_t)\right) \\
&= \frac{\mu}{2}\left[\log\det\left(\nabla^2\overline{L}_{t-1}(\mathrm{vec}(\rho_t))\right) - \log\det\left(\nabla^2\overline{L}_t(\mathrm{vec}(\rho_t))\right)\right] \\
&= \frac{\mu}{2}\left[\log\det\left(H_t - \nabla^2\overline{f}_t(\mathrm{vec}(\rho_t))\right) - \log\det\left(H_t\right)\right] \\
&= \frac{\mu}{2}\log\det\left(I - H_t^{-1/2}\nabla^2\overline{f}_t(\mathrm{vec}(\rho_t))H_t^{-1/2}\right),
\end{aligned}
$$

where strict convexity of $\overline{L}_t$ ensures the existence of $H_t^{-1/2}$. We can further simplify this representation of $\mathrm{Gain}_t$ by observing that

$$
\nabla^2\overline{f}_t(\mathrm{vec}(\rho_t)) = \frac{\mathrm{vec}(A_t)\mathrm{vec}(A_t)^*}{(\mathrm{tr}(A_t\rho))^2} = \nabla\overline{f}_t(\mathrm{vec}(\rho_t))\nabla\overline{f}_t(\mathrm{vec}(\rho_t))^*. \tag{6}
$$

Therefore, the matrix $H_t^{-1/2}\nabla^2\overline{f}_t(\mathrm{vec}(\rho_t))H_t^{-1/2}$ is single-ranked and has two distinct eigenvalues: One eigenvalue is 0 of multiplicity $(d^2 - 1)$; the other eigenvalue is

$$
\mathrm{tr}\left(H_t^{-1/2}\nabla^2\overline{f}_t(\mathrm{vec}(\rho_t))H_t^{-1/2}\right) = \langle\nabla\overline{f}_t(\mathrm{vec}(\rho_t)), H_t^{-1}\nabla\overline{f}_t(\mathrm{vec}(\rho_t))\rangle = \pi_t.
$$

of multiplicity 1. Then, the matrix $(I - H_t^{-1/2}\nabla^2\overline{f}_t(\mathrm{vec}(\rho_t))H_t^{-1/2})$ has eigenvalue 1 of multiplicity $(d^2 - 1)$ and $(1 - \pi_t)$ of multiplicity 1 and hence we obtain

$$
\mathrm{Gain}_t = \frac{\mu}{2}\log(1 - \pi_t).
$$

By Lemma 33 and the fact that $\log(1 - x) \leq -x$ for all $0 \leq x < 1$, we obtain

$$
\mathrm{Gain}_t \leq -\frac{\mu}{2}\pi_t.
$$

### 3.2.3. BOUNDING $\mathrm{Miss}_t$

Define

$$
\underline{P}_t(\rho) := L_t(\rho) + \mu\left(V_t(\rho_t) + \langle\nabla\overline{V}_t(\mathrm{vec}(\rho_t), \rho - \rho_t)\rangle\right).
$$

By Corollary 12, the function $V_t$ is convex. Consequently, $\underline{P}_t(\rho) \leq P_t(\rho)$, and we have

$$
\mathrm{Miss}_t = P_t(\rho_t) - P_t(\rho_{t+1}) \leq \underline{P}_t(\rho_t) - \underline{P}_t(\rho_{t+1}),
$$

where we have used the fact that $P_t(\rho_t) = \underline{P}_t(\rho_t)$.

The following lemma allows us to bypass the affine reparameterization step in the regret analysis by Jézéquel et al. (2022). Its proof is analogous to that in Tsai et al. (2023b, Lemma D.1) and is deferred to Appendix E (Lemma 34).

**Lemma 14** *It holds that*

$$
\underline{P}_t(\rho_t) - \underline{P}_t(\rho_{t+1}) \leq \left\|\nabla\overline{f}_t(\mathrm{vec}(\rho_t)) + \mu\left(\nabla\overline{V}_t(\mathrm{vec}(\rho_t)) - \nabla\overline{V}_{t-1}(\mathrm{vec}(\rho_t))\right)\right\|_{H_t^{-1}}^2
$$

*if*

$$
\left\|\nabla\overline{f}_t(\mathrm{vec}(\rho_t)) + \mu\left(\nabla\overline{V}_t(\mathrm{vec}(\rho_t)) - \nabla\overline{V}_{t-1}(\mathrm{vec}(\rho_t))\right)\right\|_{H_t^{-1}} \leq \frac{1}{2}.
$$

It remains to show that

$$\left\|\nabla \overline{f}_t(\text{vec}(\rho_t)) + \mu \left(\nabla \overline{V}_t(\text{vec}(\rho_t)) - \nabla \overline{V}_{t-1}(\text{vec}(\rho_t)))\right\|_{H_t^{-1}} \leq \min\left\{\frac{1}{2}, \frac{\mu}{2}\pi_t\right\}. \tag{7}$$

Similarly as in Section 3.2.2, we have

$$V_t(\rho_t) - V_{t-1}(\rho_t) = -\frac{1}{2}\log(1 - \pi_t(\rho_t)).$$

A tedious calculation then gives

$$\nabla \overline{V}_t(\text{vec}(\rho_t)) - \nabla \overline{V}_t(\text{vec}(\rho_t)) = \frac{\pi_t \nabla \overline{f}_t(\text{vec}(\rho_t)) + (1/2)w_t}{1 - \pi_t},$$

where we define

$$w_t := \frac{\nabla \overline{\varphi}_t(\text{vec}(\rho_t))}{\text{tr}(A_t \rho_t)^2},$$

$$\varphi_t(\rho) := \|\text{vec}(A_t)\|_{H_t(\rho)^{-1}}^2. \tag{8}$$

By the triangle inequality, we obtain

$$\left\|\nabla \overline{f}_t(\text{vec}(\rho_t)) + \mu \left(\nabla \overline{V}_t(\text{vec}(\rho_t)) - \nabla \overline{V}_{t-1}(\text{vec}(\rho_t)))\right\|_{H_t^{-1}}$$
$$\leq \frac{1}{(1-\pi_t)^2}\left[(1 + (\mu - 1)\pi_t)\sqrt{\pi_t} + \frac{\mu}{2}\|w_t\|_{H_t^{-1}}\right]^2. \tag{9}$$

It remains to bound $\pi_t$ and $\|w_t\|_{H_t^{-1}}$. By Lemma 33, we have $\pi_t \leq 1/(\lambda + 1)$. The following lemma bounds $\|w_t\|_{H_t^{-1}}$. Its proof is deferred to Appendix E (Lemma 35).

**Lemma 15** *It holds that* $\|w_t\|_{H_t^{-1}} \leq 2\pi_t$.

Plugging the upper bounds of $\pi_t$ and $\|w_t\|_{H_t^{-1}}$ into the inequality (9), we write

$$\left\|\nabla \overline{f}_t(\text{vec}(\rho_t)) + \mu \left(\nabla \overline{V}_t(\text{vec}(\rho_t)) - \nabla \overline{V}_{t-1}(\text{vec}(\rho_t)))\right\|_{H_t^{-1}}$$
$$\leq \frac{1}{(1-\pi_t)^2}\left[(1 + (\mu - 1)\pi_t)\sqrt{\pi_t} + \mu\pi_t\right]^2$$
$$\leq \left(\frac{\lambda + 1}{\lambda}\right)^2\left[(1 + (\mu - 1)\pi_t)\sqrt{\pi_t} + \mu\pi_t\right]^2$$
$$\leq \left(\frac{\lambda + 1}{\lambda}\right)^2\left(1 + \frac{\mu - 1}{\lambda + 1} + \frac{\mu}{\sqrt{\lambda + 1}}\right)^2 \pi_t$$
$$\leq \frac{\lambda + 1}{\lambda^2}\left(1 + \frac{\mu - 1}{\lambda + 1} + \frac{\mu}{\sqrt{\lambda + 1}}\right)^2.$$

It is then easily checked that the desired inequality (7) holds with $\lambda = 300$ and $\mu = 10$.

## Acknowledgments

We thank the anonymous reviewers for their valuable suggestions. This work is supported by the Young Scholar Fellowship (Einstein Program) of the National Science and Technology Council (NSTC) of Taiwan under grant number NSTC 112-2636-E-002-003; the 2030 Cross-Generation Young Scholars Program (Excellent Young Scholars) of the NSTC under grant number NSTC 112-2628-E-002-019-MY3; the research project "Geometry of Quantum Learning and Optimization" of National Taiwan University under grant number NTU-CC-114L895006; and the Academic Career Development Research Program (Laurel Research Project) of National Taiwan University under grant numbers NTU-CDP-113L7763 and NTU-CDP-114L7744.

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

## Appendix A. Convex Analysis and Calculus with Density Matrices

Notice that the action sets for Physicist in this work and previous ones (Aaronson et al., 2018, 2019; Bansal et al., 2024; Chen et al., 2022; Yang et al., 2020) are all subsets of the set of Hermitian complex matrices. Therefore, the loss functions are defined on complex variables. If we adopt the complex analysis perspective, extend the domains of the loss functions, and view them as general

complex-valued functions of complex matrices, then two subtle issues arise. First, the field of complex numbers $\mathbb{C}$ is not ordered, so existing inequalities for real-valued convex functions does not directly apply. Second, we may require the loss functions to be holomorphic (complex-differentiable) to adopt existing strategies for regret analyses.

Let $f$ be the loss function considered in this or any of the previous works. A simple solution is to consider the function $\varphi_{x,v}(t) := f(x + tv)$ for any Hermitian matrices $x, v$. It is easily checked that the function $\varphi$ is always real-valued, and hence is a function from $\mathbb{R}$ to $\mathbb{R}$. The convexity of $f$ then follows from those of $\varphi_{x,v}$ for any Hermitian matrices $x$ and $v$; standard convex analysis results for $f$ follows from the corresponding ones for $\varphi_{x,v}$.

We formalize the calculus notions associated with the above understanding below; the reader is referred to the book of Bauschke and Combettes (2017) for further details. Notice that the set of Hermitian matrices with the Hilbert-Schmidt inner product forms a finite-dimensional real Hilbert space. Let $\mathbb{H}$ be any finite-dimensional real Hilbert space. We say the function $f$ is (Gâteaux) differentiable at a point $x \in \mathbb{H}$ if there exists a linear mapping $Df(x) : \mathbb{H} \to \mathbb{R}$ such that

$$Df(x)[v] = \lim_{\alpha \downarrow 0} \frac{f(x + \alpha v) - f(x)}{\alpha}, \quad \forall v \in \mathbb{H};$$

then, the gradient of $f$ at $x$ is defined as the unique vector $\nabla f(x) \in \mathbb{H}$ satisfying

$$Df(x)[v] = \langle \nabla f(x), v \rangle, \quad \forall v \in \mathbb{H}.$$

Similarly, we say that function $f$ is twice differentiable at a point $x \in \mathbb{H}$ if there exists a linear mapping $D^2 f(x) : \mathbb{H} \to \mathbb{H}$ such that

$$D^2 f(x)[v] = \lim_{\alpha \downarrow 0} \frac{Df(x + \alpha v) - Df(x)}{\alpha}, \quad \forall v \in \mathbb{H};$$

then, the Hessian of $f$ at $x$ is defined as the unique linear mapping $\nabla^2 f(x) : \mathbb{H} \to \mathbb{H}$ such that

$$D^2 f(x)[v, v] = \langle v, \nabla^2 f(x)v \rangle, \quad \forall v \in \mathbb{H}.$$

Then, standard results for convex functions on Euclidean spaces apply, such as the following.

**Theorem 16 (Bauschke and Combettes (2017, Proposition 17.7))** *Let* $f : \mathbb{H} \to (-\infty, \infty]$ *be proper for some real Hilbert space* $\mathbb{H}$. *Suppose that* $\operatorname{dom} f$ *is open and convex, and that* $f$ *is differentiable on* $\operatorname{dom} f$. *Then, the following are equivalent.*

- *The function $f$ is convex.*

- *For any $x, y \in \operatorname{dom} f$, $f(y) \geq f(x) + \langle \nabla f(x), y - x \rangle$.*

- *For any $x, y \in \operatorname{dom} f$, $\langle \nabla f(y) - \nabla f(x), y - x \rangle \geq 0$.*

*If in addition, $f$ is twice differentiable on $\operatorname{dom} f$, then each of the above is equivalent to the following.*

- *For any $x \in \operatorname{dom} f$ and $v \in \mathbb{H}$, $\langle v, \nabla^2 f(x)v \rangle \geq 0$.*

It will be convenient to identify $\nabla^2 f$ with a complex matrix, as we do in Lemma 27. Let $\mathcal{H}^d$ be the real Hilbert space of $d \times d$ Hermitian matrices with the Hilbert-Schmidt inner product $\langle \cdot, \cdot \rangle_{\text{HS}}$. For any $x, v, u \in \mathcal{H}^d$, denote by $\text{vec}(x), \text{vec}(v), \text{vec}(u) \in \mathbb{C}^{d^2}$ the vectorizations of the matrices $x, v, u \in \mathbb{C}^{d \times d}$, we have

$$D^2 f(x)[v, u] = \langle v, \nabla^2 f(x) u \rangle_{\text{HS}} = \langle \text{vec}(v), \nabla^2 \overline{f}(\text{vec}(x)) \cdot \text{vec}(u) \rangle$$

where $\langle \cdot, \cdot \rangle$ is the standard inner product in $\mathbb{C}^{d^2}$ and $\nabla^2 \overline{f}(\text{vec}(x))$ is the unique $d^2 \times d^2$ complex matrix satisfying this identity.

The following lemma provides the chain rule for the Gâteaux derivatives. We will adopt it to compute the derivatives of the volumetric barrier.

**Lemma 17 (Bauschke and Combettes (2017, Fact 2.51))** *Let $\mathcal{H}_1, \mathcal{H}_2, \mathcal{H}_3$ be real Hilbert spaces. Let $f : \mathcal{H}_1 \to \mathcal{H}_2$ and $g : \mathcal{H}_2 \to \mathcal{H}_3$ be Gâteaux differentiable functions. Then, the composite function $f \circ g$ is Gâteaux differentiable and the directional derivative of it is given by*

$$D(f \circ g)(x)[v] = Df(g(x)) \left[ Dg(x)[v] \right]$$

*for all $x, v \in \mathcal{H}_1$.*

## Appendix B. Self-Concordant Functions

The following facts about self-concordant functions, necessary for proving Theorem 1, can be found in Nesterov's textbook (Nesterov, 2018). Notice that Nesterov (2018) already has already discussed the theory of self-concordant functions within the framework of finite-dimensional real Hilbert spaces. Denote by $\mathbb{H}$ a real Hilbert space.

**Definition 18** *A function $f : \text{dom } f \to \mathbb{R}$ with an open domain is $M_f$-self-concordant for some $M_f > 0$ if $f \in C^3(\text{dom } f)$ and for all $x \in \text{dom} f, u \in \mathbb{H}$, we have*

$$\left| D^3 f(x)[u, u, u] \right| \leq 2 M_f \langle u, \nabla^2 f(x) u \rangle^{3/2}.$$

**Lemma 19** *A function $f \in C^3(\text{dom } f)$ is $M_f$-self-concordant for some $M_f > 0$ if and only if*

$$\left| D^3 f(x)[u, v, w] \right| \leq 2 M_f \|u\|_{\nabla^2 f(x)} \|v\|_{\nabla^2 f(x)} \|w\|_{\nabla^2 f(x)}$$

*for all $x \in \text{dom } f$ and $u, v, w \in \mathbb{H}$.*

The weighted sum of two self-concordant functions is also a self-concordant function.

**Lemma 20 ((Nesterov, 2018, Theorem 5.1.1))** *Let $f_1$ be an $M_1$-self-concordant function and $f_2$ be an $M_2$-self-concordant function. For some $\alpha, \beta > 0$, the function $f(x) = \alpha f_1(x) + \beta f_2(x)$ is $M$-self-concordant with*

$$M = \max \left\{ \frac{1}{\sqrt{\alpha}} M_1, \frac{1}{\sqrt{\beta}} M_2 \right\}.$$

Let $f$ be a self-concordant function. Define the local norm

$$\|v\|_{\nabla^2 f(x)} := \sqrt{\langle v, \nabla^2 f(x)v \rangle}, \quad \forall x \in \text{dom } f.$$

The following lemma demonstrates that self-concordant functions possess properties that resemble those of strongly convex functions.

**Lemma 21 ((Nesterov, 2018, Theorem 5.1.8))** *Let $f$ be an $M_f$-self-concordant function. Then, for all $x, y \in \text{dom } f$,*

$$\langle \nabla f(y) - \nabla f(x), y - x \rangle \geq \frac{\|y - x\|_{\nabla^2 f(x)}^2}{1 + \|y - x\|_{\nabla^2 f(x)}}$$

*and*

$$f(y) \geq f(x) + \langle \nabla f(x), y - x \rangle + \frac{1}{M_f^2} \omega(M_f \parallel y - x \parallel_{\nabla^2 f(x)}),$$

*where $\omega(t) := t - \log(1 + t)$.*

We will need the following properties of the function $\omega$.

**Lemma 22 ((Nesterov, 2018, Lemma 5.1.5))**

1. *The Fenchel conjugate of $\omega$ is given by $\omega_*(t) := -t - \log(1 - t)$.*

2. *For any $t > 0$, we have*

$$\frac{t^2}{2(1 + t)} \leq \omega(t) \leq \frac{t^2}{2 + t}.$$

3. *For any $t \in [0, 1)$, we have*

$$\frac{t^2}{2 - t} \leq \omega_*(t) \leq \frac{t^2}{2(1 - t)}.$$

We then show that the vectorized loss functions $\overline{f}_t$ in LL-OLQS and the vectorized log-determinant function are all 1-self-concordant.

**Lemma 23** *The vectorized loss functions $\overline{f}_t$ in LL-OLQS are 1-self-concordant.*

**Proof** This follows from the 1-self-concordance of the logarithmic function and the affine invariance of self-concordance (Nesterov, 2018, Example 5.1.1 and Theorem 5.12). ∎

**Lemma 24** *Let $R \colon \mathcal{H}^d \to \mathbb{R}$ be the log-determinant function, i.e., $R(\rho) = -\log \det \rho$. Then, for any integer $n \geq 1$,*

$$D^n R(\rho)[V_1, \ldots, V_n] = \frac{(-1)^n}{n} \sum_{\sigma \in S_n} \text{tr} \left( \rho^{-1} V_{\sigma(1)} \rho^{-1} V_{\sigma(2)} \cdots \rho^{-1} V_{\sigma(n)} \right)$$

*where the sum is over all permutations $\sigma$ of $\{1, \ldots, n\}$. In particular,*

$$DR(\rho)[U] = -\text{tr} \left( \rho^{-1} U \right)$$
$$D^2 R(\rho)[U, V] = \text{tr} \left( \rho^{-1} U \rho^{-1} V \right)$$

**Proof** The expression of $DR(\rho)[U]$ follows from Nesterov (2018, Lemma 5.4.6); the proof therein considers real matrices but direct extends for the complex case. Let $f\colon \mathcal{H}^d \to \mathcal{H}^d$ be the map $f(\rho) = \rho^{-1}$, we have (Bhatia, 1997, Example X.4.2 (iii))

$$Df(\rho)[U] = -\rho^{-1}U\rho^{-1}.$$

Thereby, we obtain the expression of $D^2R(\rho)[U,U]$. The general expression of $D^nR[V_1,\ldots,V_n]$ follows from the product rule for differentiation and induction on $n$. ∎

**Lemma 25** *Let $R$ be the log-determinant function, i.e., $R(\rho) = -\log\det\rho$. Then, the vectorized regularization function $\overline{R}$ is $1$-self-concordant.*

**Proof** This lemma is the vectorized version of Nesterov (2018, Lemma 5.4.3). Lemma 24 gives

$$D^3\overline{R}(\mathrm{vec}(\rho))[\mathrm{vec}(U),\mathrm{vec}(U),\mathrm{vec}(U)] = D^3R(\rho)[U,U,U] = 2\,\mathrm{tr}\left(U\rho^{-1}U\rho^{-1}U\rho^{-1}\right),$$
$$D^2\overline{R}(\mathrm{vec}(\rho))[\mathrm{vec}(U),\mathrm{vec}(U)] = D^2R(\rho)[U,U] = \mathrm{tr}\left(U\rho^{-1}U\rho^{-1}\right).$$

Denote the eigenvalues of the matrix $\rho^{-1/2}U\rho^{-1/2}$ as $\lambda_1,\ldots,\lambda_{d^2}$, which may not be all distinct. Then, it suffices to show that

$$\left|\sum_{i=1}^{d^2}\lambda_i^3\right| \le 2\left(\sum_{i=1}^{d^2}\lambda_i^2\right)^{3/2}.$$

By the Cauchy-Schwarz inequality, we write

$$\left|\sum_{i=1}^{d^2}\lambda_i^3\right|^2 \le \left(\sum_{i=1}^{d^2}|\lambda_i|^3\right)^2$$
$$= \left(\sum_{i=1}^{d^2}|\lambda_i||\lambda_i|^2\right)^2$$
$$\le \left(\sum_{i=1}^{d^2}|\lambda_i|^2\right)\left(\sum_{i=1}^{d^2}|\lambda_i|^4\right)$$
$$\le \left(\sum_{i=1}^{d^2}|\lambda_i|^2\right)\left(\sum_{i=1}^{d^2}|\lambda_i|^2\right)^2$$
$$= \left(\sum_{i=1}^{d^2}|\lambda_i|^2\right)^3,$$

which proves the lemma. ∎

## Appendix C. Properties of Kronecker Product

Let $A$ and $B$ be $m \times n$ and $k \times l$ matrices. The *Kronecker product* of $A$ and $B$, denoted by $A \otimes B$, is defined as the following $mk \times nl$ block matrix

$$
\begin{pmatrix}
A_{11}B & \cdots & A_{1n}B \\
\vdots & & \vdots \\
A_{m1}B & \cdots & A_{mn}B
\end{pmatrix}.
$$

.

We will use the following properties of Kronecker products in Lemmas 27 and 28 to compute the explicit forms of higher-order derivatives of $-\log \det(\rho)$.

**Lemma 26** *Let $A, B, C$ and $D$ be $d \times d$ matrices. Then,*

1. $(A \otimes B)(C \otimes D) = AC \otimes BD$

2. $(A \otimes B)^* = A^* \otimes B^*$

3. *If $A \geq 0$ and $B \geq 0$, then $A \otimes B \geq 0$.*

4. *(Magnus and Neudecker, 2019, Theorem 18.5)* $\text{vec}(ABC) = (C^\mathsf{T} \otimes A) \text{vec}(B)$.

**Lemma 27** *Let $R \colon \mathcal{H}^d \to \mathbb{R}$ be the negative log-determinant function, i.e., $R(\rho) = -\log \det \rho$. Then,*

$$
\nabla^2 \overline{R}(\text{vec}(\rho)) = (\rho^{-1})^\mathsf{T} \otimes \rho^{-1}
$$

**Proof** For all $V, W \in \mathcal{H}^d$, by Lemma 24 and Lemma 26, we write

$$
\begin{aligned}
\langle \text{vec}(V), \nabla^2 \overline{R}(\text{vec}(\rho)) \text{vec}(W) \rangle &= D^2 \overline{R}(\rho)[\text{vec}V, \text{vec}W] \\
&= \text{tr}(V \rho^{-1} W \rho^{-1}) \\
&= \langle \text{vec}(V), \text{vec}(\rho^{-1} W \rho^{-1}) \rangle \\
&= \langle \text{vec}(V), ((\rho^{-1})^\mathsf{T} \otimes \rho^{-1}) \text{vec}(W) \rangle
\end{aligned}
$$

for all $U \in \mathcal{H}^d$ ∎

**Lemma 28** *Let $R \colon \mathcal{H}^d \to \mathbb{R}$ be the negative log-determinant function, i.e., $R(\rho) = -\log \det \rho$. Let $U \in \mathcal{H}^d$ and $u = \text{vec}(U)$, Then,*

$$
D^3 \overline{R}(\text{vec}(\rho))[u] = -(\rho^{-1}U\rho^{-1})^\mathsf{T} \otimes \rho^{-1} - (\rho^{-1})^\mathsf{T} \otimes (\rho^{-1}U\rho^{-1}).
$$

**Proof** For all $V, W \in \mathcal{H}^d$, let $v = \text{vec}(V), w = \text{vec}(W)$. By Lemma 24,

$$
D^3 \overline{R}(\text{vec}(\rho))[u, v, w] = -\text{tr}(\rho^{-1}U\rho^{-1}V\rho^{-1}W) - \text{tr}(\rho^{-1}U\rho^{-1}W\rho^{-1}V)
$$

.

By Lemma 26 we have

$$
\begin{aligned}
&\langle \mathrm{vec}(V), D^3 \overline{R}(\mathrm{vec}(\rho))[u]\mathrm{vec}(W) \rangle \\
={}& D^3 R(\rho)[u, v, w] \\
={}& - \mathrm{tr}\left( \rho^{-1}U\rho^{-1}V\rho^{-1}W \right) - \mathrm{tr}\left( \rho^{-1}U\rho^{-1}W\rho^{-1}V \right) \\
={}& \langle \mathrm{vec}V, \mathrm{vec}\left( -\rho^{-1}W\rho^{-1}U\rho^{-1} - \rho^{-1}U\rho^{-1}W\rho^{-1} \right) \rangle \\
={}& \langle \mathrm{vec}(V), \left( -(\rho^{-1}U\rho^{-1})^{\mathsf{T}} \otimes \rho^{-1} - (\rho^{-1})^{\mathsf{T}} \otimes (\rho^{-1}U\rho^{-1}) \right) \mathrm{vec}(W) \rangle
\end{aligned}
$$

∎

## Appendix D.  Missing Proofs in Section 2

### D.1.  Proof of Lemma 5

By the Cauchy-Schwarz inequality and VB-convexity of $\varphi_1$ and $\varphi_2$,

$$
\begin{aligned}
&D^4\varphi(x)[u, u, v, v]D^2\varphi(x)[v, v] \\
&= \left( \alpha D^4\varphi_1(x)[u, u, v, v] + \beta D^4\varphi_2(x)[u, u, v, v] \right) \left( \alpha D^2\varphi_1(x)[v, v] + \beta D^2\varphi_2(x)[v, v] \right) \\
&\geq \left( \alpha\sqrt{D^4\varphi_1(x)[u, u, v, v]D^2\varphi_1(x)[v, v]} + \beta\sqrt{D^4\varphi_1(x)[u, u, v, v]D^2\varphi_2(x)[v, v]} \right)^2 \\
&\geq \left( \alpha\sqrt{\frac{3}{2}}D^3\varphi_1(x)[u, v, v] + \beta\sqrt{\frac{3}{2}}D^3\varphi_2(x)[u, v, v] \right)^2 \\
&= \frac{3}{2}\left( D^3\varphi(x)[u, v, v] \right)^2 .
\end{aligned}
$$

### D.2.  Proof of Lemma 6

Let $v = B^{-1/2}u$. Then, we have

$$
\langle u, B^{-1/2}AB^{-1/2}u \rangle \langle u, u \rangle \geq \langle u, B^{-1/2}CB^{-1/2}u \rangle^2 .
$$

Denote the eigendecomposition of $B^{-1/2}CB^{-1/2}$ as $\sum_{i=1}^{d} \lambda_i u_i u_i^*$, where $\lambda_i$ are eigenvalues and $u_i$ are the associated eigenvectors. Then the set $\{u_i\}$ forms an orthonormal basis and we have

$I = \sum_i u_i u_i^*$. We write

$$\text{tr}(AB^{-1}) = \text{tr}\left(B^{-1/2}AB^{-1/2}\right)$$

$$= \text{tr}\left(B^{-1/2}AB^{-1/2}\sum_{i=1}^{n} u_i u_i^*\right)$$

$$= \sum_{i=1}^{n} \langle u_i, B^{-1/2}AB^{-1/2}u_i\rangle \langle u_i, u_i\rangle$$

$$\geq \sum_{i=1}^{n} \langle u_i, B^{-1/2}CB^{-1/2}u_i\rangle^2$$

$$= \sum_{i=1}^{n} \lambda_i^2$$

$$= \text{tr}\left(B^{-1/2}CB^{-1/2}B^{-1/2}CB^{-1/2}\right).$$

The inequality follows from the fact that trace is invariant under cyclic shifts.

## Appendix E.  Lemmas in Regret Analysis (Section 3.2)

All notations are defined as in Section 3.

Tsai et al. (2023a, Proposition 7) proved that the functions $R - f_t$ in LL-OLQS are convex on $\mathcal{D}_d$. Below is a vectorized analogue of their result.

**Lemma 29**  *It holds that, for all positive definite $\rho \in \mathcal{D}_d$,*

$$\nabla^2 \overline{f}_t(\text{vec}(\rho)) \leq \nabla^2 \overline{R}(\text{vec}(\rho)).$$

**Proof**

A direct calculation, as well as Lemma 27, gives

$$\nabla^2 \overline{f}_t(\text{vec}(\rho)) = \frac{\text{vec}(A_t)\text{vec}(A_t)^*}{(\text{tr}(A_t\rho))^2}, \quad \nabla^2 \overline{R}(\text{vec}(\rho)) = (\rho^{-1})^\intercal \otimes \rho^{-1}.$$

We desire to prove that $\langle m, \nabla^2 \overline{f}_t(\text{vec}(\rho))m\rangle \leq \langle m, \nabla^2 \overline{R}(\text{vec}(\rho))m\rangle$ for all vectors $m \in \mathbb{C}^{d^2}$. Define $M = \text{vec}^{-1}(m)$ for any vector $v$. Then, we desire to prove that the inequality

$$\frac{(\text{tr}(A_t M))^2}{(\text{tr}(A_t\rho))^2} \leq \langle m, \text{vec}(\rho^{-1}M\rho^{-1})\rangle = \text{tr}(M\rho^{-1}M\rho^{-1})$$

holds for all $m \in \mathbb{R}^{d^2}$, where we have used the fact that $(A \otimes B)\text{vec}(V) = \text{vec}(BVA^\intercal)$ for any matrices $A$, $B$, and $V$. Let $\langle\cdot,\cdot\rangle_{\text{HS}}$ denotes the Hilbert-Schmidt inner product. By the Cauchy-Schwarz inequality and the fact that $\text{tr}(A^2) \leq (\text{tr}(A))^2$ for any positive semi-definite matrix $A$, we

write

$$
\begin{aligned}
\mathrm{tr}\big(M\rho^{-1}M\rho^{-1}\big) &= \langle \rho^{-1/2}M\rho^{-1/2}, \rho^{-1/2}M\rho^{-1/2}\rangle_{\mathrm{HS}} \\
&\geq \frac{\langle \rho^{1/2}A_t\rho^{1/2}, \rho^{-1/2}M\rho^{-1/2}\rangle_{\mathrm{HS}}^2}{\langle \rho^{1/2}A_t\rho^{1/2}, \rho^{1/2}A_t\rho^{1/2}\rangle_{\mathrm{HS}}} \\
&= \frac{(\mathrm{tr}(A_t M))^2}{\big(\mathrm{tr}\big(\rho^{1/2}A_t\rho^{1/2}\rho^{1/2}A_t\rho^{1/2}\big)\big)} \\
&\geq \frac{(\mathrm{tr}(A_t M))^2}{\big(\mathrm{tr}\big(\rho^{1/2}A_t\rho^{1/2}\big)\big)^2} \\
&= \frac{(\mathrm{tr}(A_t M))^2}{(\mathrm{tr}(A_t\rho))^2}.
\end{aligned}
$$

This proves the lemma. ∎

**Lemma 30** *Let $\rho_T^\star \in \mathrm{argmin}_{\rho \in \mathcal{D}_d} L_T(\rho)$. Then, it holds that*

$$
\rho_T^\star \geq \frac{\lambda}{T + \lambda d}I.
$$

**Proof** By the optimality condition,

$$
\mathrm{tr}\left[\left(\lambda(\rho_T^\star)^{-1} + \sum_{t=1}^T \frac{A_t}{\mathrm{tr}\big(A_t\rho_T^\star\big)}\right)(\rho - \rho_T^\star)\right] \leq 0, \quad \forall \rho \in \mathcal{D}_d.
$$

Then, we write

$$
\lambda\, \mathrm{tr}\big(\rho(\rho_T^\star)^{-1}\big) \leq \lambda d + \sum_{t=1}^T \frac{\mathrm{tr}(A_t(\rho_T^\star - \rho))}{\mathrm{tr}\big(A_t\rho_T^\star\big)} \leq \lambda d + T, \quad \forall \rho \in \mathcal{D}_d.
$$

Taking maximum of the left-hand side over all $\rho \in \mathcal{D}_d$, we obtain that the largest eigenvalue of $(\rho_T^\star)^{-1}$ is bounded from above by $(\lambda d + T)/\lambda$. This implies that the smallest eigenvalue of $\rho_T^\star$ is bounded from below by $\lambda/(\lambda d + T)$. ∎

The following lemma is a direct generalization of (Jézéquel et al., 2022, Lemma A.2) for the quantum setup. We omit the proof as it is almost the same as that for the classical case.

**Lemma 31** *For any $\rho \in \mathcal{D}_d$ and $\alpha \in (0,1)$, define $[\rho]_\alpha := (1-\alpha\rho) + (\alpha/d)I$. Then, it holds that*

$$
f_t([\rho]_\alpha) - f_t(\rho) \leq \frac{\alpha}{1-\alpha}.
$$

The following lemma is necessary to bounding $\pi_t$.

**Lemma 32** *Let $H \in \mathbb{C}^{d\times d}$ be a positive definite matrix. Then, for any $v \in \mathbb{C}^d$ and $\gamma > 0$, it holds that $vv^* \leq \gamma H$ if and only if $\|v\|_{H^{-1}} \leq \sqrt{\gamma}$.*

**Proof** Notice that the dual norm of $\|\cdot\|_H$ is $\|\cdot\|_{H^{-1}}$ and vice versa. If $vv^* \leq \gamma H$, then we have

$$\|v\|_{H^{-1}}^2 = \max_{u \neq 0} \frac{|\langle u, v \rangle|^2}{\|u\|_H^2} = \max_{u \neq 0} \frac{\langle u, vv^*u \rangle}{\|u\|_H^2} \leq \gamma.$$

On the other hand, if $\|v\|_{H^{-1}} \leq \sqrt{\gamma}$, then for all $u \in \mathbb{R}^d$, we have

$$\langle u, \gamma H u \rangle = \gamma \|u\|_H^2 \geq \gamma \frac{|\langle v, u \rangle|^2}{\|v\|_{H^{-1}}^2} \geq |\langle v, u \rangle|^2 = \langle u, vv^*u \rangle, \quad \forall v \in \mathbb{C}^d, v \neq 0.$$

∎

**Lemma 33** *It holds that*

$$\pi_t \leq 1/(\lambda + 1) < 1.$$

**Proof** By the equation (6) and Lemma 29, we write

$$(\lambda + 1)\nabla \overline{f}_t(\mathrm{vec}(\rho_t))\nabla \overline{f}_t(\mathrm{vec}(\rho_t))^* = (\lambda + 1)\nabla^2 \overline{f}_t(\mathrm{vec}(\rho_t)) \tag{10}$$

$$\leq \lambda \nabla^2 \overline{R}(\mathrm{vec}(\rho_t)) + \nabla^2 \overline{f}_t(\mathrm{vec}(\rho_t)) \tag{11}$$

$$\leq H_t. \tag{12}$$

By Lemma 32, we obtain

$$\pi_t = \left\|\nabla \overline{f}_t(\mathrm{vec}(\rho_t))\right\|_{H_t^{-1}}^2 \leq \frac{1}{\lambda + 1}.$$

∎

**Lemma 34** *It holds that*

$$\underline{P}_t(\rho_t) - \underline{P}_t(\rho_{t+1}) \leq \left\|\nabla \overline{f}_t(\mathrm{vec}(\rho_t)) + \mu \left(\nabla \overline{V}_t(\mathrm{vec}(\rho_t)) - \nabla \overline{V}_{t-1}(\mathrm{vec}(\rho_t))\right)\right\|_{H_t^{-1}}^2$$

*if*

$$\left\|\nabla \overline{f}_t(\mathrm{vec}(\rho_t)) + \mu \left(\nabla \overline{V}_t(\mathrm{vec}(\rho_t)) - \nabla \overline{V}_{t-1}(\mathrm{vec}(\rho_t))\right)\right\|_{H_t^{-1}} \leq \frac{1}{2}.$$

**Proof** By Lemma 23, Lemma 25, and Lemma 20, the function $\underline{P}_t$ after vectorization is 1-self-concordant. By Lemma 21, we write

$$\underline{P}_t(\rho_t) - \underline{P}_t(\rho_{t+1}) \leq \langle \nabla \underline{P}_t, \mathrm{vec}(\rho_t) - \mathrm{vec}(\rho_{t+1}) \rangle - \omega\left(\|\mathrm{vec}(\rho_t) - \mathrm{vec}(\rho_{t+1})\|_{H_t}\right). \tag{13}$$

By the optimality condition for $\rho_t$, we have

$$\begin{aligned}
&\langle \nabla \overline{P}_{t-1}(\rho_t), \mathrm{vec}(\rho_{t+1}) - \mathrm{vec}(\rho_t) \rangle \\
&= \langle \nabla \overline{L}_{t-1}(\mathrm{vec}(\rho_t)) + \mu \nabla \overline{V}_{t-1}(\mathrm{vec}(\rho_t)), \mathrm{vec}(\rho_{t+1}) - \mathrm{vec}(\rho_t) \rangle \\
&\geq 0, \quad \forall \rho \in \mathcal{D}_d.
\end{aligned}$$

Then, we write

$$\langle \nabla \underline{P}_t, \mathrm{vec}(\rho_t) - \mathrm{vec}(\rho_{t+1}) \rangle$$
$$= \langle \nabla \overline{L}_t(\mathrm{vec}(\rho)) + \mu \nabla \overline{V}_t(\mathrm{vec}(\rho_t)), \mathrm{vec}(\rho_t) - \mathrm{vec}(\rho_{t+1}) \rangle$$
$$\leq \langle \nabla \overline{f}_t(\mathrm{vec}(\rho_t)) + \mu \nabla \overline{V}_t(\mathrm{vec}(\rho_t)) - \mu \nabla \overline{V}_{t-1}(\rho_t), \mathrm{vec}(\rho_t) - \mathrm{vec}(\rho_{t+1}) \rangle$$
$$\leq \left\| \nabla \overline{f}_t(\mathrm{vec}(\rho_t)) + \mu \nabla \overline{V}_t(\mathrm{vec}(\rho_t)) - \mu \nabla \overline{V}_{t-1}(\rho_t) \right\|_{H_t^{-1}} \| \mathrm{vec}(\rho_t) - \mathrm{vec}(\rho_{t+1}) \|_{H_t}. \quad (14)$$

where the last inequality holds because $\| \cdot \|_{H_t}$ and $\| \cdot \|_{H_t^{-1}}$ are dual norms. Combining the two inequalities (13) and (14), we obtain

$$\underline{P}_t(\rho_t) - \underline{P}_t(\rho_{t+1})$$
$$\leq \left\| \nabla \overline{f}_t(\mathrm{vec}(\rho_t)) + \mu \nabla \overline{V}_t(\mathrm{vec}(\rho_t)) - \mu \nabla \overline{V}_{t-1}(\rho_t) \right\|_{H_t^{-1}} \| \mathrm{vec}(\rho_t) - \mathrm{vec}(\rho_{t+1}) \|_{H_t}$$
$$- \omega \left( \| \mathrm{vec}(\rho_t) - \mathrm{vec}(\rho_{t+1}) \|_{H_t} \right).$$

By the definition of the Fenchel conjugate, we have

$$\underline{P}_t(\rho_t) - \underline{P}_t(\rho_{t+1}) \leq \omega_* \left( \left\| \nabla \overline{f}_t(\mathrm{vec}(\rho_t)) + \mu \nabla \overline{V}_t(\mathrm{vec}(\rho_t)) - \mu \nabla \overline{V}_{t-1}(\rho_t) \right\|_{H_t^{-1}} \right).$$

The lemma follows from Lemma 22. ∎

**Lemma 35** *It holds that* $\| w_t \|_{H_t^{-1}} \leq 2\pi_t$.

**Proof** For any $u \in \mathcal{H}^d$, by Lemma 24 and the chain rule (Lemma 17), we write

$$|\langle w_t, u \rangle| = \left| \frac{D\varphi(\rho_t)[u]}{\mathrm{tr}(A_t \rho_t)^2} \right|$$
$$= \left| \frac{- \mathrm{tr}\left( H_t^{-1} \mathrm{vec}(A_t) \mathrm{vec}(A_t)^* H_t^{-1} DH_t(\rho_t)[u] \right)}{\mathrm{tr}(A_t \rho_t)^2} \right|$$
$$= \left| \frac{-D^3 \overline{L}_t(\mathrm{vec}(\rho_t))[u, H_t^{-1} \mathrm{vec}(A_t), H_t^{-1} \mathrm{vec}(A_t)]}{\mathrm{tr}(A_t \rho_t)^2} \right|.$$

where the function $\varphi_t$ is given in Section 3.2.3 (8). Given that

$$\nabla \overline{f}_t(\mathrm{vec}(\rho_t)) = \frac{-\mathrm{vec}(A_t)}{\mathrm{tr}(A_t \rho_t)},$$

we obtain

$$|\langle w_t, u \rangle| = \left| -D^3 \overline{L}_t \left( \mathrm{vec}(\rho_t) \right) \left[ u, H_t^{-1} \nabla \overline{f}_t(\mathrm{vec}(\rho_t)), H_t^{-1} \nabla \overline{f}_t(\mathrm{vec}(\rho_t)) \right] \right|.$$

Lemma 23 and Lemma 25 have established that the vectorized loss functions $\overline{f}_t$ and vectorized regularizer $\overline{R}$ are all 1-self-concordant. Hence, the function $\overline{L}_t$ is also 1-self-concordant by Lemma 20. By Lemma 19, we write

$$|\langle w_t, u \rangle| \leq 2 \| u \|_{H_t} \left\| H_t^{-1} \nabla \overline{f}_t(\mathrm{vec}(\rho_t)) \right\|_{H_t}^2 = 2\pi_t \| u \|_{H_t},$$

which implies that

$$w_t w_t^* \leq 4\pi_t^2 H_t.$$

It remains to apply Lemma 32. ∎

## Appendix F. Implementation of VB-FTRL for LL-OLQS in Section 3.1

We use the same notation as in Section 3.1. To implement VB-FTRL for LL-OLQS, we need to solve the following optimization problem at the $t + 1$-th round,

$$\rho_{t+1} \in \underset{\rho \in \mathcal{D}_d}{\operatorname{argmin}} P_t(\rho). \tag{15}$$

where

$$P_t(\rho) = L_t(\rho) + \mu V_t(\rho),$$

$$L_t(\rho) = \sum_{\tau=1}^{t} f_\tau(\rho) + \lambda R(\rho),$$

$$V_t(\rho) := \frac{1}{2} \log \det \nabla^2 \overline{L}_t(\operatorname{vec}(\rho)),$$

$$f_\tau(\rho) = -\log \operatorname{tr}(A_\tau \rho)$$

$$R(\rho) = -\log \det(\rho).$$

It is easily checked that the function $L_t$ is convex. By Corollary 8 and the VB-convexity of $\overline{L}_t$ (Corollary 12), the volumetric barrier $V_t(\rho)$ is convex, so the optimization problem (15) is convex. This allows us to apply standard algorithms, such as cutting plane methods, to solve the convex optimization problem (15). In particular, we summarize the ellipsoid method and its convergence guarantee in Theorem 36. We then specialize the ellipsoid method for implementing VB-FTRL and analyze the per-round computational complexity in Theorem 38.

### F.1. Review of Ellipsoid Method

Let $\mathcal{H}$ be an $d'$-dimensional real Hilbert space. Consider the problem of minimizing a convex differentiable function $f$ over a non-empty closed convex set $\mathcal{X} \subset \mathcal{X}$. The ellipsoid method proceeds as follows.

- Let the initial ellipsoid be

$$E_1 = \left\{ x \in \mathcal{H} \mid \langle x - x_1, H_1^{-1}(x - x_1) \rangle \leq 1 \right\},$$

  for some $x_1 \in \mathcal{H}$ and positive definite matrix $H_1$ such that $\mathcal{X} \subseteq E_1$.

- For every $k \geq 1$, compute

$$x_{k+1} = x_k - \frac{1}{d' + 1} \frac{H_k g_k}{\sqrt{g_k^* H_k g_k}},$$

$$H_{k+1} = \frac{d'^2}{d'^2 - 1} \left( H_k - \frac{2}{d' + 1} \frac{H_k g_k g_k^* H_k}{g_k^* H_k g_k} \right),$$

  for some $g_k$ satisfying

$$\langle g_k, x - x_k \rangle \leq 0, \quad \forall x \in \underset{x \in \mathcal{X}}{\operatorname{argmin}} f(x).$$

**Theorem 36 (Lee and Vempala (2025, Lemma 3.3))** *Define*

$$E_k := \{\, x \in \mathcal{H} \mid \langle x - x_k, H_k^{-1}(x - x_k) \rangle \le 1 \,\}.$$

*The ellipsoid method achieves the following:*

- *The ellipsoids $E_k$ contain the minimizer $x_\star$.*

- *The volume of $E_k$ decays at a linear rate:*

$$\mathrm{vol}(E_{K+1}) \le \mathrm{e}^{-K/(2d'+2)} \,\mathrm{vol}(E_1).$$

### F.2. Specializing Ellipsoid Method for VB-FTRL

We now detail the implementation of the ellipsoid method for solving the optimization problem (15). In our case, $\mathcal{H}$ is the space of vectorized $d \times d$ Hermitian matrices equipped with the standard complex inner product; the set $\mathcal{X}$ corresponds to the set of vectorized $d \times d$ density matrices $\mathcal{D}$; and the function $f$ to be minimized is $\overline{P}_t$. Since the set of extreme points of $\mathcal{X}$ coincides with the set of rank-one projection matrices and has unit radius, we have $\mathrm{vol}(E_1) = 1$. Thus, Theorem 36 implies that the ellipsoid algorithm identifies an ellipsoid of volume $\varepsilon$ containing the minimizer in $O(d^2 \log(1/\varepsilon))$ iterations.

Then, we discuss how to set the vectors $g_k$ in the ellipsoid method.

- If $\mathrm{vec}^{-1}(x_k)$ is positive definite, then the optimality condition ensures that setting $g_k = \nabla \overline{P}_t(x_k)$ suffices.

- Otherwise, $\nabla \overline{P}_t(x_k)$ is not well defined. Nevertheless, it is easily checked that setting $g_k = -\mathrm{vec}(vv^*)$, where $v$ is an eigenvector of $\mathrm{vec}^{-1}(x_k)$ corresponding to a negative eigenvalue, suffices.

Indeed, for the first case, it is unnecessary to compute $\nabla \overline{P}_t(\mathrm{vec}^{-1}(x_k))$. Let $H_k = \sum_j \lambda_j u_j u_j^*$ be the eigendecomposition of $H_k$. Define $d_j := D\overline{P}_t(x_k)(u_j)$. We can write the iteration rule equivalently as

$$x_{k+1} = x_k - \sum_{j=1}^{d^2} \left( \frac{1}{d^2 + 1} \frac{\lambda_j d_j}{\sqrt{\sum_{j=1}^{d^2} \lambda_j d_j^2}} u_j \right),$$

$$H_{k+1} = \frac{d^4}{d^4 - 1} \left( H_k - \frac{2}{d^2 + 1} \frac{\sum_{j=1}^{d^2} (\lambda_j^2 d_j^2 u_j u_j^*)}{\sum_{j=1}^{d^2} \lambda_j d_j^2} \right),$$

showing it suffices to compute the directional derivatives of $\overline{P}_t$. Below we provide the explicit formula of the directional derivative.

**Lemma 37** *For any Hermitian positive semi-definite matrix $\rho \in \mathbb{C}^{d \times d}$ and $U \in \mathbb{C}^{d \times d}$, the directional derivative of $\overline{P}_t$ is given by:*

$$D\overline{P}_t(\mathrm{vec}(\rho))[\mathrm{vec}(U)]$$

$$= -\sum_{\tau=1}^{t} \frac{\mathrm{tr}(A_\tau U)}{\mathrm{tr}(A_\tau \rho)} + \lambda \,\mathrm{tr}(\rho^{-1} U) + 2\mu \sum_{\tau=1}^{t} \frac{\mathrm{tr}(A_\tau U)}{(\mathrm{tr}(A_\tau \rho))^2} \|\mathrm{vec}(A_\tau)\|_{\nabla^{-2} \overline{L}_t(\mathrm{vec}(\rho))}$$

$$- \frac{\lambda \mu}{2} \,\mathrm{tr}\left( \nabla^{-2} \overline{L}_t(\mathrm{vec}(\rho)) \left( (\rho^{-1} U \rho^{-1})^\intercal \otimes \rho^{-1} + (\rho^{-1})^\intercal \otimes (\rho^{-1} U \rho^{-1}) \right) \right),$$

*where*

$$\nabla^2 \overline{L}_t(\mathrm{vec}(\rho)) = \sum_{\tau=1}^{t} \frac{1}{(\mathrm{tr}(A_\tau \rho))^2} \mathrm{vec}(A_\tau) \mathrm{vec}(A_\tau)^* + \lambda \left[ (\rho^{-1})^\top \otimes \rho^{-1} \right]. \tag{16}$$

**Proof** Recall that $\overline{P}_t = \overline{L}_t + \mu \overline{V}_t$. We derive the directional derivatives of $\overline{L}_t$ and $\overline{V}_t$ separately. A direct calculation gives

$$\nabla \overline{L_t}(\mathrm{vec}(\rho))[\mathrm{vec}(U)] = -\sum_{\tau=1}^{t} \frac{\mathrm{tr}(A_\tau U)}{\mathrm{tr}(A_\tau \rho)} + \lambda \, \mathrm{tr}(\rho^{-1} U). \tag{17}$$

Applying the chain rule (Lemma 17), we obtain

$$D\overline{V_t}(\mathrm{vec}(\rho))[\mathrm{vec}(U)] = \frac{1}{2} \mathrm{tr}\left( \nabla^{-2} \overline{L}_t(\mathrm{vec}(\rho)) D^3 \overline{L}_t (\mathrm{vec}(\rho)) [\mathrm{vec}(U)] \right). \tag{18}$$

A computation similar to the one in the proof of Lemma 29 gives the formula (16). By Lemma 28, we have

$$D^3 \overline{L}_t (\mathrm{vec}(\rho)) [\mathrm{vec}(U)]$$
$$= \sum_{\tau=1}^{t} -2 \frac{\mathrm{tr}(A_\tau U)}{(\mathrm{tr}(A_\tau \rho))^3} \mathrm{vec}(A_\tau) \mathrm{vec}(A_\tau)^* + \lambda D^3 \overline{R} (\mathrm{vec}(\rho)) [\mathrm{vec}(U)]$$
$$= \sum_{\tau=1}^{t} 2 \frac{\mathrm{tr}(A_\tau U)}{(\mathrm{tr}(A_\tau \rho))^2} \mathrm{vec}(A_\tau) \mathrm{vec}(A_\tau)^* + \lambda \left( -(\rho^{-1} U \rho^{-1})^\top \otimes \rho^{-1} - (\rho^{-1})^\top \otimes (\rho^{-1} U \rho^{-1}) \right). \tag{19}$$

The lemma follows by combining the equations (17), (18), and (19).

∎

Finally, we analyze the per-round computational complexity of VB-FTRL for LL-OLQS as described in Section 3.1.

**Theorem 38** *VB-FTRL for LL-OLQS can be implemented with a per-round computational complexity of $O(Td^8 \log(1/\varepsilon) + d^{10} \log(1/\varepsilon))$.*

**Proof** The computational complexity of computing the trace of the product of two $d^2 \times d^2$ matrices is $O(d^4)$. The computational complexity of computing the inverse of a $d^2 \times d^2$ matrix is $O(d^6)$. Therefore, computing each $d_j$ at the $t$-th round takes $O(td^4 + d^6)$ time, and one iteration of the ellipsoid method at the $t$-th round takes $O(td^6 + d^8)$ time. Recall that the iteration complexity of the ellipsoid method is $O(d^2 \log(1/\varepsilon))$. Thus, the per-round computational complexity of VB-FTRL for LL-OLQS is $O(Td^8 \log(1/\varepsilon) + d^{10} \log(1/\varepsilon))$. ∎

