# OpenReview forum: "Online Learning of Quantum States with Logarithmic Loss via VB-FTRL"
_algorithmiclearningtheory.org/ALT/2025/Conference — ALT 2025_

### Official Review · Reviewer_ubiv · 2024-11-09

**Rating:** 7
**Confidence:** 3

**Review:**

This paper studies the problem of Online learning of quantum states with respect to the Logarithmic loss function. In general, in online learning framework, there are two players: say the Physicist and the Reality. The game is played for $T$ rounds in a sequential manner. At every round (say at round $t$), the Physicist decides a density matrix $\rho_t$, and then Reality announces a Hermitian positive semi-definite matrix $A_t$, and the Physicist suffers a loss of $- \log tr(A_t \rho)$. The primary goal is to design efficient algorithm to minimize the cumulative loss of the Physicist over the $T$ rounds against all possible strategies of the Reality. This is commonly referred as the regret minimization problem and has been extensively studied for several decades in the classical setting.

Recently Aaronson et al. (NeurIPS 2018) initiated the study of online learning in the quantum setting, and since then there have been several works in this setting. However, this problem has been primarily studied when the loss function is convex and Lipschitz continuous. In this work, the authors studied the problem for the logarithmic loss function which they define as LL-OLQS (Online learning of Quantum states under logarithmic loss). However, since the logarithmic loss function is not Lipschitz continuous, the known techniques do not extend directly.

In this setting, the current best result is by Zimmert et al. (COLT 22) which achieves the regret of $O(d^2 \log T)$, where $d$ is the dimension. However, the algorithm of Zimmert et al. evaluates a high dimensional integral in every round. The authors in this paper design an algorithm for LL-OLQS with regret $O(d^2 \log(d+T))$ where each iteration of the game can be performed in polynomial time using semidefinite programming.

The authors follow the techniques of Jézéquel et al. (2022) who designed an algorithm VB-FTRL for the celebrated Online Portfolio Selection (OPS) problem. However, the approach of Jézéquel et al. does not immediately translate to the quantum setting. In particular, the technique of affine parametrization does not work here. Moreover, Jézéquel et al. used the convexity of the Volumetric Barrier (VB) (see Section 2, paragraph 1) for bounding the final regret, which does not directly translate to the quantum setting.

In order to bypass the second bottleneck, the authors define the notions of VB-convexity in the quantum context (see Definition 3 & Theorem 7, Section 2.1). They show that the volumetric barrier associated with any convex function remains convex (Lemma 5 & Corollary 8), which is crucially used for the regret analysis in Section 3. To bypass the first bottleneck, they used techniques from the work of Tsai et al. (NeurIPS 23) (see Lemma 13, Section 3).

The proof of the main theorem (Theorem 1) is described in Section 3, which follows the Follow the Regularized Leader (FTRL) approach from Jézéquel et al. with the new notion of the Volumetric barrier convexity and establishes the total regret of $O(d^2 \log(d+T))$. There is a gap of $d$ with the current lower bound of $d \log T$ which follows from the universal portfolio selection. The authors also mention this in the introduction.

I think the new notion of the volumetric barrier (VB) convexity in this context is novel and would be of interest to the ALT community. I support accepting the paper.

Comments:

1. I think adding a table of prior results mentioned in the introduction would be better to follow.
2. I think presenting a brief overview of the regret analysis in Section 3 would improve the readability of the paper.

**Paper Award:**

No

---

> ### Author Response · Authors · 2024-11-24
>
> We thank you for appreciating this work.
>
> - On adding a table of prior results: For online portfolio selection, a table summarizing prior work can be found in the paper by Jézéquel et al. (2022), which is up to date to the best of our knowledge. Regarding the quantum generalization with logarithmic loss, we are only aware of four existing works: Exponential Weight (a direct generalization of Universal Portfolio), Schrodinger’s BISONS, Q-Soft-Bayes, and Q-LB-OMD. Hence, we believe that a paragraph suffices to summarize them. If you think adding a table would be beneficial, please let us know, and we will include it in the revision.
> - On providing brief overview of the regret analysis: Sure. We will provide it.

---

> > ### Comment · Reviewer_ubiv · 2024-11-25
> >
> > Thanks for your response. I am keeping my rating.

---

### Official Review · Reviewer_b7RH · 2024-11-09
**Nice result, but the presentation can be improved**

**Rating:** 7
**Confidence:** 2

**Review:**

The paper studies online learning of quantum states with logarithmic loss (LL-OLQS). They extend the VB-FTRL algorithm by Jezequel et al. (2022) from the online portfolio setting to the LL-OLQS setting. This results in an polynomial-time algorithm achieving d^2 log(d+T) regret. This gives a different solution than Zimmert et al. (2022)'s universal portfolio and BISONS, improving the first's computational complexity and the second's regret bound.

I believe the result is novel and the analysis is sound based on my own reading. However, I hope that more high-level explanation and the ideas behind the proof could be spent in the main text, instead of just laying out the proofs, even if the ideas may already be in the previous work of Jezequel et al. (2022).  For example, what's the motivation behind the choice of the regularizer, and why it addresses the problem better than previous work.  In particular, I find that in your analysis there is a term "Gain_t", which induces a negative regret cancelling out "Miss_t".  On the other hand, Zimmert et al. (2022) said that they use "negative regret via linear bias", whose goal is also to cancel out another positive term.  Are these terms in your paper and their paper related to each other? What's the key to improve their bound? Also, it is mentioned that your analysis "avoid the affine reparametrization step" in Jezequel et al. (2022). What's the implication of this? Does it simplify the proof, or improve the bound (i.e., if you don't use it you will get worse bound)? Please clarify.

Besides, what is the more precise per-round complexity of your algorithm? It is only stated as "the complexity of solving a semidefinite program", but this doesn't seem to be sufficient for a fair comparison with previous work (like universal portfolio).

Besides, I would prefer the paper to provide more intermediate steps of calculation, rather than saying "a direct calculation gives..." and omitting details. For example, at the bottom of Page 7.  It will just make the proofs easier to follow.

Below are minor points:
- "VB" always appears in the abbreviated form. It should be spelled out at the first time
-  Typo: Lemma 11  "LB-convex"-->"VB-convex"

**Paper Award:**

No

---

> ### Author Response · Authors · 2024-11-24
>
> We thank you for the careful review and for appreciating this work.
>
> - On the choice of the regularizer: We refer to Section 2 of the paper by Jezequel et al. (2022), which provides an interesting argument to motivate the regularizer, via approximating Cover’s Universal Portfolio using Gaussian distributions.
> - On "Gain_t" and "Miss_t": We refer to Section 3.1 of the paper by Jezequel et al. (2022) for a brief description of the analysis strategy. We will provide a brief overview of the regret analysis in the revision.
> - On the relation with the work of Zimmert et al. (2022): The starting point for Zimmert et al. (2022) was to improve the algorithm proposed by Luo et al. (2018). In contrast, our algorithm is a generalization of Jezequel et al. (2022). We do not notice any clear connection between the works of Zimmert et al. (2022) and Jezequel et al. (2022).
> - On reparametrizaton: Jezequel et al. (2022) reparametrized the probability simplex in $R^d$ to the set $\lbrace x \in \mathbb{R}^{d - 1} : x_i \geq 0, \sum_i x_i \leq 1 \rbrace$ in $\mathbb{R}^{d - 1}$. The reason is to have an action set with a non-empty interior in order to bound a Newton decrement by textbook results (Section 3.1, Jezequel et al., 2022). For the quantum setup, however, it is unclear to us how to find an analogous reparametrization of the set of quantum density matrices. Therefore, circumventing the reparametrization step is essential to our analysis.
> - On per-round complexity: Please see our response to Reviewer m5x2 above.
> - On providing more calculation details: The skipped steps are rather direct applications of the matrix calculus results in the appendix and the chain rule. In the revision, we will provide clear pointers to relevant lemmas in the appendix.
>
> **Reference.**
> H. Luo, C.-Y. Wei, and K. Zheng. Efficient online portfolio with logarithmic regret. *Adv. Neural Information Processing Systems 31 (NeurIPS 2018)*. 2018.

---

### Official Review · Reviewer_m5x2 · 2024-11-09
**An interesting extension of the result of Jezequel, Ostrovskii, and Gaillard**

**Rating:** 8
**Confidence:** 4

**Review:**

The authors study the quantum version of Cover’s portfolio problem. While it is known that matrix multiplicative weights yield a regret bound of d^2 log(T) for this problem, this method lacks an explicit implementation and requires the computation of complex multidimensional integrals. Existing algorithms in the literature give polynomial-time computation but have suboptimal regret bounds. The authors build on recent work by Jezequel, Ostrovskii, and Gaillard to achieve an almost optimal regret of d^2 log(d + T) via semi-definite programming.

One potential weakness of the paper is that it is not clear what the computation time is. Although only semi-definite programming is required, the polynomial dependence in the quantum case could be significant. It would be useful to compare this with the mentioned complexity bounds of O(d^3) per round, which, while suboptimal in terms of regret, offer more straightforward computation.

Although the analysis follows several steps outlined in the work by Jezequel, Ostrovskii, and Gaillard, the paper introduces technical tricks and distinctions, which are clearly highlighted. I liked that contributions are presented transparently and without unnecessary claims. While I have not checked the proofs in depth, the overall context suggests the results and analysis are solid. The topic is very relevant, and the paper deserves to be accepted.

**Paper Award:**

No

---

> ### Author Response · Authors · 2024-11-24
>
> We thank your for appreciating this work.
>
> - On the per-round computation time: Regarding the per-round computation time, consider the optimization problem of computing the iterate in a round. By Lemma 3.2 in the book draft *Techniques in Optimization and Sampling* by Lee and Vempala (see also Lemma 2.3 of the book *Convex Optimization: Algorithms and Complexity* by Bubeck), the ellipsoid method would require $O(d^2 \log (1 / \varepsilon))$ iterations to find a set of volume $\varepsilon$ containing the minimizer. Computing the gradient in each iteration of the ellipsoid method takes $O ( T d^6 )$ time. Consequently, the per-round computation time is $O ( T d^{8} \log ( 1 / \varepsilon ) )$. We will detail this in the revision.

---

> > ### Comment · Reviewer_m5x2 · 2024-12-01
> >
> > Thank you for the reply. Please, mention this in your revision. I keep my score.

---

### Meta-Review · Area_Chair_XciH · 2024-12-05

**Recommendation:** Accept
**Confidence:** 5

**Metareview:**

This paper provides a new algorithm for the online learning of quantum states with log loss problem. The algorithm is based on the VB-FTRL of Jézéquel et al. (2022), which was designed for the online portfolio selection problem. The extension to the quantum setting involves several non-trivial challenges, not least of which is extending the necessary convex analysis to the complex domain for density matrices. For this, the authors introduce a new tool, VB-convexity, which is very intriguing and could be of independent interest. The result is an algorithm that provides (a possibly minimax optimal) regret bound of O(d^2 log(d+T)), with poly(d, T) running time per iteration.

Reviewers were unanimous in their approval of the paper as a good fit for ALT, so I am happy to recommend acceptance. In the next version, it would be helpful if the authors could provide more intuition for the analysis in the main text, as one of the reviewers suggests. It would be also be nice (but not required) if the authors could reflect some more on the notion of VB-convexity, and whether it has connections to self-concordance, given that the definitions seems so related, because it seems this could be a very useful tool for other work.

**Paper Award:**

No